# Are Demanding Job Situations Associated with Alcohol-Related Presenteeism? The WIRUS-Screening Study

**DOI:** 10.3390/ijerph18116169

**Published:** 2021-06-07

**Authors:** Tore Bonsaksen, Mikkel Magnus Thørrisen, Jens Christoffer Skogen, Morten Hesse, Randi Wågø Aas

**Affiliations:** 1Department of Health and Nursing Science, Inland University of Applied Sciences, 2418 Elverum, Norway; 2Faculty of Health Studies, VID Specialized University, 4306 Sandnes, Norway; 3Department of Occupational Therapy, Oslo Metropolitan University, 0130 Oslo, Norway; mithor@oslomet.no; 4Department of Public Health, University of Stavanger, 4036 Stavanger, Norway; jens.christoffer.skogen@fhi.no; 5Center for Alcohol & Drug Research, Stavanger University Hospital, 4068 Stavanger, Norway; 6Department of Health Promotion, Norwegian Institute of Public Health, 5015 Bergen, Norway; 7Center for Alcohol and Drug Research, Aarhus University, 2300 Copenhagen, Denmark; mh.crf@psy.au.dk

**Keywords:** alcohol, effort-reward imbalance, health promotion, job content questionnaire, presenteeism, psychosocial work environment, sick leave, work performance, workplace

## Abstract

Alcohol-related presenteeism (impaired work performance caused by alcohol use) is an important but under-researched topic. The aim of this study was to explore whether psychosocial work environment factors were associated with alcohol-related presenteeism. A cross sectional study of Norwegian employees (*n* = 6620) was conducted. Logistic regression analyses were used for estimating associations with alcohol-related presenteeism, which was reported among 473 (7.1%) of the employees. Adjusted by age, gender, education level and managerial level, higher levels of overcommitment to work were associated with alcohol-related presenteeism. Higher age, male gender and higher education were also associated with alcohol-related presenteeism. Occupational health services and employers should especially focus on overcommitted employees when designing workplace health promotion programs. Modifying attitudes towards alcohol-related presenteeism among overcommitted employees may be of importance for safety at work.

## 1. Introduction

In an organizational perspective, productivity in the workplace is reduced when employees are on sick leave and absent from work, but also when they come to work with reduced capacity, known as presenteeism. Presenteeism has been defined as “*decreased on-the-job performance due to the presence of health problems*” [1], and has been shown to incur even greater costs than absenteeism [2]. Employees suffering from chronic disease such as allergies, arthritis, chronic pain, diabetes, gastro-intestinal conditions, musculoskeletal problems, mental illnesses are likely to have higher levels of presenteeism than their generally healthy colleagues. In addition, health risks (e.g., overweight, physical inactivity, higher alcohol consumption) have been associated with higher levels of presenteeism [1,3,4].

In addition to individual health factors, aspects of the work environment and the employees’ attitudes toward their job may also contribute to presenteeism [5,6,7,8]. Several studies have been guided by the Job Demand Control (JDC) [9] and the Effort Reward Imbalance (ERI) [10] models in their design and measurement. Karasek’s JDC model proposes that job demands, the employee’s decision latitude and the perceived support from managers and colleagues interact and constitute the psychosocial work environment [9]. Importantly, high job demands are not viewed as problematic per se but may be problematic in cases where the employee’s decision latitude is low; shaping what is known as a high-strain job. Siegrist, on the other hand, conceptualized a healthy work environment as a state of balance between the employee’s work efforts and their perceived rewards [10]. An adverse imbalance between the two factors occurs when high efforts are perceived as insufficiently rewarded. To achieve desired rewards, employees may overcommit themselves to work, while this strategy in turn may increase the perceived effort-reward imbalance. 

In support of the reasoning derived from the JDC and ERI models, studies have shown presenteeism to be associated with higher job demands, control, support, efforts and commitment to work, and lower levels of rewards [6,11,12,13,14]. Employees with low-strain jobs (high control combined with low demands) have been shown to have lower odds of presenteeism, compared to employees with passive jobs, active jobs and high-strain jobs [14]. Associations between work environment variables and presenteeism have also been found to be mediated by psychological distress [15]. However, findings in relation to presenteeism and measures of control and support have not been consistent, as non-existent or oppositely directed associations have also been reported [11,12,14,16]. The lack of congruence may indicate that control and support may serve as resources via two different routes: they may render it possible to work with reduced pace and effort, but also to be absent from work when needed. 

In western populations, drinking alcohol is common while it also represents a major public health threat, with disability and a range of diseases attributable to it [17]. Higher levels of alcohol consumption, and binge drinking in particular, is also a risk factor for presenteeism, as suggested from previous studies and reviews [18,19,20]. Specifically, in one study of Norwegian employees, 11% reported reduced work performance due to drinking the previous day during the last year [21]. While generally not well tolerated among Norwegian employees, attitudes toward presenteeism due to alcohol use appear to be more liberal among employees who have own experience with presenteeism and among employees with lower education levels [22]. However, more restrictive attitudes among those with higher education may not always translate into coherent practices, as the risk of alcohol-related presenteeism has been shown to be higher in people with higher education and income levels [23].

In the current study, alcohol-related presenteeism is conceptualized as reduced work functioning resulting from alcohol consumption. It is a specific subtype of presenteeism; a product of a relationship between alcohol consumption and impaired work performance [18]. Thus, impaired work performance and its causal attribution to drinking alcohol is inherent in the concept of presenteeism as used in this study. In view of the previous studies, the JDC and ERI models appear to be frequently used—and to be useful—for understanding presenteeism. It is reasonable to assume that some of the factors previously found to be associated with presenteeism (e.g., commitment to work) can be associated also with alcohol-related presenteeism. However, one may also predict oppositely directed associations, compared to earlier results. For example, while high support from supervisors and colleagues may increase presenteeism in the case of most health problems, it may not carry over to what may be coined self-imposed impairments, such as those resulting from alcohol use. Further, while high work demands generally make employees more inclined to attend the job, even when unable to perform as usual, adverse reactions from managers and colleagues alike may occur in cases where alcohol use is the cause for the impaired work performance. In view of the widespread use of alcohol in the population and its high potential for reducing productivity, safety and well-being in the workplace, a better understanding of alcohol-related presenteeism is warranted.

The aim of this study was to explore associations between alcohol-related presenteeism and: (i) levels of psychological demands, decision latitude and support, (ii) levels of perceived effort, reward and overcommitment at work, (iii) perceived effort-reward imbalance, (iv) a high-strain job, low perceived support, high effort-reward imbalance and high overcommitment, and (v) accumulated work environment risk factors.

## 2. Methods

### 2.1. Design

This study is part of the ongoing Norwegian national Workplace Interventions preventing Risky Use of alcohol and Sick leave (WIRUS) project. The present study used cross-sectional data from employees in the private and public Norwegian workforce. In collaboration with the addiction competence environment KoRus Stavanger and the University of Stavanger, companies were recruited by convenience through three occupational health services (OHS) in Norway. Twenty-two companies served by the three OHS agreed to participate and provided e-mail addresses for all their employees. Provided they satisfied the inclusion criteria (see Section 2.2), all of these employees were invited to participate in the study. The recruitment strategy sought to gather a heterogeneous sample of employees and workplaces. Hence, the companies were recruited based on geographical, sector and industry diversity, representing the following economic activities: Transportation/storage, education, manufacturing, public administration, human health/social work activities, and accommodation/food service. Given that the recruitment challenges in the large-scaled project involved both the enterprise level and the individual level, the recruitment was extended over a period of several years, with new places of work and new employees added each year (2014–2019).

### 2.2. Sample and Inclusion Criteria

Employees were invited to participate in a web-based alcohol screening study, which entailed completing questionnaires related to alcohol-related presenteeism, psychosocial work environment, and a range of individual background variables. The inclusion criteria were: (1) over 18 years of age, (2) being employed in a private or public enterprise where data collection was agreed upon by management, and (3) provided informed consent to participate. Altogether 30,811 individuals from 22 enterprises were asked to participate in the study. Of these, 8542 (response rate 27.7%) completed the questionnaires.

In addition, to be included in the present study, the participants were required to have valid responses to all relevant variables (see Measures section). After removal of individuals with incomplete data, the final sample was constituted of 6620 (4617 female and 2003 male) employees from 22 companies (21.5% of the eligible participants). The mean age was 45.0 years (standard deviation [SD] = 11.3, range: 19 to 71). Of the employees, 78% had higher education and 80% were regular employees. 

### 2.3. Measures

#### 2.3.1. Sociodemographic Variables

Of sociodemographic variables, the study included age (range 18–71 years), gender, education level (primary/lower secondary school; upper secondary/high school; university/college ≤ 4 years; university/college > 4 years), and occupational level (regular employee, middle management, top executive, or other). In the logistic regression analysis, age was recoded to reflect ten-year age bands; education level was recoded to reflect high school or lower vs. higher education; and managerial responsibility level was dichotomized into regular employee versus managers, the latter category collapsing the middle management and top executive levels into one category.

#### 2.3.2. Job Demand, Control and Support

Relevant items taken from the Job Content Questionnaire (JCQ) [9] were used to measure the psychosocial work environment. Decision latitude, a concept comprising the employee’s own control over decisions in the workplace (decision authority), and the possibility of developing and using personal skills in the job (skill discretion), was measured with the sum score of the nine relevant items (Cronbach’s α = 0.76). Psychosocial demands of the job, such as having an unreasonably great workload or not having enough time to get the work done, was measured with the sum score of five items (Cronbach’s α = 0.76). Lastly, social support was measured with the sum score of eight items, comprising support from co-workers and support from the supervisor (Cronbach’s α = 0.89). Four items on the JCQ with reversed scoring were recoded prior to analysis.

#### 2.3.3. Effort-Reward Imbalance and Overcommitment

Four sub-scales derived from the effort-reward model (effort, reward, over-commitment, and effort-reward ratio) were assessed with the 16 items short version of the Effort-Reward Imbalance Questionnaire [24]. Five items with reversed scoring were recoded prior to analysis. The sum score of three items constitute the scale ‘effort’ (Cronbach’s α = 0.74), while the sum score of seven items constituted the scales ‘reward’ (Cronbach’s α = 0.62), and a sum score of six items constituted the ‘over-commitment’ scale (Cronbach’s α = 0.82). The sub-scale effort-reward imbalance (ERI) ratio was calculated by the formula: ERI = effort/reward × *c*, where *c* represents a correction factor that adjusts for the unequal number of items on the two scales. When using the short version of the instrument, the correction factor is 7/3 [25]. ERI measures approaching zero indicate much perceived reward in relation to the efforts spent, while ERI measures exceeding ‘1’ indicate a perceived imbalance between high efforts and low rewards. Employees were classified as having high ERI and high overcommitment in cases where their scores were higher than the sample median value on the relevant variables. Previous studies have shown good psychometric properties related to each of the sub-scales [26].

#### 2.3.4. Work Environment Risk Factors

In accordance with Karasek and colleagues, categories of psychosocial work environment are constructed by combining the decision latitude and psychosocial demands variables [9]. Employees were classified as having a high-strain job provided their score on psychological demand was higher than the median value and their score on decision latitude was lower than the median value. Employees were classified as having low support if their score was lower than the median value in the sample.

#### 2.3.5. Cumulative Risk Variable

A cumulative risk variable was constructed by summarizing the risk factors (high-strain job, low support, high ERI, and high overcommitment) relevant for each employee. Scores on this variable ranged 0–4, with higher scores indicating higher accumulated risk. 

#### 2.3.6. Risky Drinking

The presence or absence of risky drinking was measured with the Alcohol Use Disorders Identification Test (AUDIT) [27]. The AUDIT consists of ten items (Cronbach’s α = 0.73) designed to measure alcohol consumption (three items), alcohol dependence (three items), and alcohol-related problems (four items). Each item was scored 0–4, resulting in a sum score ranging from 0 to 40. Risky drinking was conceptualized as a drinking pattern that may lead to “*social, medical, domestic, job and financial problems*” ([27], p. 7), and was operationalized as a sum score of 8 or higher on the AUDIT (absence of risky drinking = 0 [AUDIT 0–7]; presence of risky drinking = 1 [AUDIT 8–40]). Application of this threshold has been found to represent a satisfactory compromise between sensitivity and specificity [27]. 

#### 2.3.7. Alcohol-Related Presenteeism

The outcome variable of the study was measured with one item taken from the Work Productivity and Activity Impairment Questionnaire (WPAI) [28]. The item was phrased: “While working during the last seven days, how much did your alcohol consumption impact on your productivity?” Participants were prompted to think back on days when the extent or type of work they could do was limited, or days when they achieved less than they wanted, or days they were less diligent than usual. Ratings were made on an 11-point scale, with anchoring statements at each end of the scale (0 indicating that “my alcohol consumption had no effect on my work”, while 10 indicating that “my alcohol consumption completely prevented me from working”).

### 2.4. Data Analysis

The sample was described using means (*M*) and standard deviation (*SD*) for continuous variables and using frequency (*n*) and proportion (%) for categorical variables. Due to the distribution of scores on the WPAI, this outcome variable was recoded into a variable with two categories, representing no impact on productivity (0) versus some impact on productivity (1). Differences between participants with and without alcohol-related presenteeism were investigated using independent *t*-tests and Chi Square tests for continuous and categorical variables, respectively. 

After controlling that the requirements for logistic regression analysis were met, a series of logistic regression analyses were conducted to assess associations with presenteeism. First, simple logistic regressions were conducted to estimate unadjusted associations. Second, six multiple regression models were constructed, all of which included age band, gender, education level and occupational level. Thus, the included independent variables of interest represented (i) individual background variables (age band, gender, education level, and occupational level); (ii) Job Demand Control model variables (psychosocial demands, decision latitude and social support); (iii) Effort-Reward Imbalance model variables (effort, reward, and overcommitment), (iv) effort-reward imbalance ratio, (v) work environment risk factors (high-strain job, low support, high effort-reward imbalance ratio, high overcommitment), and (vi) accumulated risk (presence of 0–4 work environment risk factors). Effect sizes were reported as odds ratio (OR) with corresponding 95% confidence intervals (CI). 

In additional analyses, decision latitude was replaced with the variable ‘decision authority’, separating out the skills discretion items in order to examine whether the personal authority aspect of decision latitude was associated with alcohol-related presenteeism. Support was separated into coworker support and manager support, and the association between these variables and alcohol-related presenteeism were tested in subsequent analyses. We also tested the association between overcommitment and alcohol-related presenteeism, while removing effort and reward from the model. In order to examine whether the relationships between significant predictors of alcohol-related presenteeism differed depending on the presence or absence of risky drinking, we conducted a series of moderator analyses. In each case, we examined the associations between the interaction terms (i.e., high AUDIT score × age; gender; education level; managerial responsibility; or overcommitment, respectively) and alcohol-related presenteeism.

## 3. Results

### 3.1. Description of the Sample

The sample characteristics, and the comparisons between employees reporting presenteeism versus no presenteeism, are displayed in Table 1. The sample consisted of 143 (2.2%) top executives, 1161 (17.5%) middle management, and 5316 (80.3%) regular workers. Most of the sample reported no alcohol-related presenteeism (92.9 %). Of the 473 employees who reported any degree of presenteeism, 401 (84.8%) reported a score of “1”, indicating a small degree of presenteeism during the preceding week. In comparison to those reporting no presenteeism, employees with presenteeism were slightly older (*p* = 0.04), were more frequently men (*p* = 0.003) and reported more frequently higher education (*p* < 0.001). The mean scores on psychosocial work environment variables were not significantly different between employees with and without presenteeism. Similarly, effort and reward were not significantly different between those with and without presenteeism, whereas those who reported any degree of presenteeism had higher scores on overcommitment (*p* = 0.009). Finally, those re- porting alcohol-related presenteeism were more likely to screen positive for risky drinking (score 8 or higher) on the AUDIT (25.2%) than those who did not (10.3%, *p* < 0.001).

### 3.2. Unadjusted Associations with Alcohol-Related Presenteeism

The results from the simple logistic regressions are displayed in Table 2. Each ten-year increase in age increased the odds of reporting alcohol-related presenteeism (OR: 1.09, *p* < 0.05). Compared to men, women had lower odds of reporting presenteeism (OR: 0.74, *p* < 0.01) while participants with higher education had higher odds compared to their counterparts (OR: 1.86, *p* < 0.001). Compared to regular employees, persons in leading positions (middle managers and top executives) had higher odds of reporting some level of presenteeism (OR: 1.31, *p* < 0.05), and employees with higher overcommitment scores had increased odds of reporting presenteeism (OR: 1.04, *p* < 0.01). Similarly, treating overcommitment as a categorical variable, employees with scores above the median sample value had higher odds of reporting presenteeism (OR: 1.25, *p* < 0.05), compared to their counterparts with scores below the median value. Otherwise, none of the independent variables showed a significant association with presenteeism.

### 3.3. Adjusted Associations with Alcohol-Related Presenteeism

The results from the multiple logistic regression analyses are displayed in Table 3. The first model, including only the sociodemographic factors as independent variables, revealed that ten-year increase in age (OR: 1.10, *p* < 0.05), male gender (OR: 0.75, *p* < 0.01) and having higher education (OR: 1.89, *p* < 0.001) were significantly associated with presenteeism. These findings were reiterated in all of the five subsequent adjusted models: higher age increased the odds of alcohol-related presenteeism during the last week (all OR: 1.10, all *p* < 0.05); female gender reduced the odds (OR ranging between 0.74 and 0.75, all *p* < 0.05); and having higher education increased the odds (OR ranging between 1.85 and 1.94, all *p* < 0.001). Only high overcommitment derived from the ERI model significantly increased the odds of presenteeism, both when used as a continuous variable (OR: 1.04, *p* < 0.05) and when used as a categorical variable (OR: 1.25, *p* < 0.05). Otherwise, none of included variables were significantly associated with alcohol-related presenteeism. 

### 3.4. Additional Analyses

When replacing the ‘decision latitude’ variable with the three items comprising ‘decision authority’ (removing the ‘skill discretion’ items), and dividing support into two variables indicative of manager support and co-worker support, respectively, the results for Model 2 did not change significantly (in Table 3). Similarly, when removing effort and reward from Model 3, we found that the association between overcommitment and alcohol-related presenteeism (adjusted by age, gender, education level and managerial responsibility) was practically unchanged, both when overcommitment was used as a continuous measure (OR: 1.03, 95% CI: 1.00–1.06) and when used as a categorical measure (OR: 1.22, 95% CI: 1.01–1.48). 

We tested gender, age, education level, managerial responsibility, and high over-commitment in interaction analyses, in order to assess whether any of these differed by the presence or absence of risky drinking. None of the interaction analyses were statistically significant, i.e., the interaction terms were not significant for gender (*p* = 0.88), age (*p* = 0.87), education level (*p* = 0.67), managerial responsibility (*p* = 0.38), or high over-commitment (*p* = 0.68).

## 4. Discussion

This study aimed to explore associations between alcohol-related presenteeism and perceptions of the work environment, as derived from the JDC and ERI models, while adjusting for sociodemographic variables. In the adjusted models, only higher levels of overcommitment were associated with higher odds of self-reported alcohol-related presenteeism. Other variables describing aspects of the work environment were not associated with alcohol-related presenteeism, whereas higher age, male gender and having higher education were associated with alcohol-related presenteeism across all models.

Presenteeism was associated with overcommitment, both when overcommitment was used as a continuous scale and when used as a categorical variable (Table 3). Overcommitment is described as employees’ striving toward high achievement because of an underlying need for approval and esteem at work [10]. The ERI model prediction is therefore that high overcommitment results in low absence from work, and high presenteeism – the employee is inclined to come to work, even when he or she is not fit to do the work that is expected of them. These predictions related to associations between high overcommitment and sickness absence/sickness presenteeism have been supported in several studies [5,11,13]. In a similar vein, high overcommitment has been found to be associated with alcohol-related problems, in particular when combined with high effort-reward imbalance [29]. Thus, the detected association between overcommitment and alcohol-related presenteeism is in line with earlier studies on presenteeism, supporting the validity of the ERI model assumption related to overcommitment even when applied specifically to alcohol-related presenteeism. 

In contrast, psychological demands imposed by the job, as measured with the ‘psychological demands’ scale [9], were not associated with alcohol-related presenteeism. In combination, the results therefore indicate that attending work with reduced capacity due to drinking is tied to the employee’s internal dedication to the job, rather than the externally imposed requirements of the job. 

Beyond overcommitment, associations between alcohol-related presenteeism and JDC and ERI model concepts were not found. This is in contrast to studies of general presenteeism showing evidence of associations with risk factors derived from these models [6,11,12,13,14]. Moreover, our findings stand in contrast to previously reported associations between presenteeism and reporting a higher number of risk factors; such as having a high-strain job, which was found to increase presenteeism among employees in Korea [14]. This could represent a difference between alcohol-related presenteeism and presenteeism due to general health problems. However, we cannot rule out that other dissimilarities between studies may contribute to this difference, such as sample compositions and different cultural contexts. As generally suggested from previous studies [11,12], more support from colleagues and managers and having more control over work tasks and their pacing may allow employees to attend work with impaired productivity while sick. Thus, reduced work capacity following alcohol use may not by itself be the cause of absence from work. In fact, a recent study showed that heavy alcohol use was negatively associated with sickness absence [30]. However, in the case of alcohol-related presenteeism, the mechanisms that generally can stimulate presenteeism may also play out differently. Employees who would agree on having high job control and support in the workplace may assume that these perceived resources cannot justify impaired work performance due to alcohol use and may be inclined to stay at home when sick (e.g., experiencing hangover), instead of attending work. A range of negative consequences have been experienced by co-workers of employees with alcohol-related presenteeism [21]; such as covering for colleagues, experiencing a poorer work environment, and worrying about their own safety. Such findings may be considered evidence of co-workers’ support being within limits. In addition, or alternatively, being absent from work when affected from alcohol use may be due to a fear having one’s drinking exposed. Following this reasoning, working within physical proximity to others, such as in shared office landscapes or in jobs requiring personal meetings with colleagues or clients, may be more important predictors of alcohol-related presenteeism than psychosocial work environment factors.

The illness flexibility model may serve as a unifying theoretical framework for understanding work attendance (absence or presence) [16]. In this model, the ability to work depends on the health status or degree of loss of function, while the moderating forces of adjustment latitude, attendance requirements and attendance incentives influence how work ability translates into actual work attendance or non-attendance. Being able to adjust work tasks and the pace of work may place the employee in a position where attending work is possible, despite impairments. High attendance requirements also work in the direction of attendance, although rather owing to the prospects of negative outcomes related to non-attendance (e.g., work piling up, higher workload on colleagues, loss of job or career opportunities). In addition, attendance incentives such as perceptions of rewards associated with attending work (social and/or professional identity, self-esteem and self-actualization) increase the probability of attendance. This study of alcohol-related presenteeism indicates that overcommitment, which is likely to reflect attendance requirements, play a more significant role than perceived control of one’s own job. This is in line with previous findings concerned with presenteeism in general, demonstrating that job resources are generally weaker correlates of presenteeism than job demands [5,31]. However, as also noted by Johansson and Lundberg [16], actual attendance requirements can be difficult to disentangle from employees’ perceptions of such requirements. Regardless of standards being internal or external, they seem to increase the chances of employees attending the job even when their work capacity is hampered due to alcohol use.

Across all statistical models, higher age, male gender and having higher education were consistently associated with alcohol-related presenteeism during the last week (Table 3). These results are in contrast to the generally non-existent or weak associations between sociodemographic variables and presenteeism, which was demonstrated in a recent meta-analysis [5]. Moreover, the findings related to age and gender were different in comparison to those reported by Allemann and co-workers [3], where younger age and female gender was associated with presenteeism among hospital employees. While differences between studies in part may depend on the use of highly specific samples versus samples recruited from a variety of enterprises, the difference between presenteeism in general versus alcohol-related presenteeism may be of greater importance for the interpretation. Drinking at risky levels are more common among younger employees [32], but their alcohol use may more often result in absenteeism rather than presenteeism. As detected in the current study, those of higher age seem to be more inclined to go to work, even when their work performance is reduced by drinking. On the other hand, risky drinking is more frequent among men more than women [32], and a pattern of binge drinking has indeed been associated with more alcohol-related presenteeism [19]. These findings may support the notion that men are more inclined to attend work when their performance is reduced due to alcohol use. 

Having higher education may increase the probability of having more job control, or adjustment latitude, but this may be dependent on work sector and specific occupation. In general, presenteeism has been found to be common in the education, welfare and health sectors [31], where loyalty to and concern for vulnerable clients is considered part of the professional culture. In view of the current study results, having higher education may potentially contribute to higher levels of overcommitment, which in turn was linked with alcohol-related presenteeism. However, as the association between education and alcohol-related presenteeism was only marginally weaker when adjusting by overcommitment, there appears to be aspects of higher education levels which function independently from overcommitment; possibly those coined work incentives, that link with alcohol-related presenteeism. Thus, an association between education level and alcohol-related presenteeism, partially mediated by overcommitment, seems viable. In view of a recent study demonstrating more negative attitudes toward alcohol-related presenteeism among those with higher education levels [22], there may be reason to further investigate the relationship between alcohol-related attitudes and practices in population subgroups.

One basic question to consider concerns whether alcohol-related presenteeism is a good or a bad thing, and whether specific circumstances may influence this perception. In accordance with the illness flexibility model [16], one may argue that presenteeism represents an alternative to absenteeism. However, determining whether presenteeism is preferable over absenteeism is not straightforward. The nature of the health condition should be considered. Absenteeism is probably preferable in cases of acute contagious conditions, while the opposite can be argued in cases of more chronic non-communicable conditions [33]. Also, the nature of the job should be considered. For instance, presenteeism may be critical in jobs where optimal work performance is of utmost importance (e.g., when operating heavy machinery). Regarding alcohol-related presenteeism, one may similarly consider the nature of the impairment and the nature of the job. For instance, impaired work performance as a result of active on-the-job intoxication may be more critical than impairments due to experiencing hangover symptoms.

Finally, while respondents with risky drinking according to the AUDIT were far more likely to report alcohol-related presenteeism, other predictors did not vary by AUDIT status, and the results reported in this study therefore apply to both workers with and without a risky drinking pattern.

### Study Strengths and Limitations

The main strength of this study is the use of a large sample of employees representing a wide variety of public and private enterprises of varying sizes. Thus, we believe the results might be relevant for a larger population of employees. Moreover, the study employed the JDC and ERI questionnaires, which cover many relevant aspects of the work environment. However, as described in more detail in other publications from the WIRUS screening study [18,19,32,34,35], the employees in the study sample were, compared with the entire Norwegian workforce, older and highly educated and had female employees overrepresented. The sample was considerably more representative for Norwegian public and state sector employees, than for private sector employees. We do not have information about the proportions of participants who were permanently and temporarily employed, respectively. This is a limitation, since people with temporary and permanent positions may vary in their risk of alcohol-related presenteeism, and it is not clear in what direction.

The most important limitations are the low response rate and the cross-sectional study design which precludes causal inferences about the reported associations. There is a possibility of reversed causality. Individuals with higher alcohol-related presenteeism were, to a larger extent, individuals with higher degree of alcohol use. These individuals may be more inclined to have an unbalanced perception of their work duties and thus perceive higher levels of overcommitment.

The data collection took place over a period of five years. While such an extended period of data collection for a cross-sectional study may be rare, alcohol consumption in Norway has been stable during the whole period [36], and there is little reason to believe that levels of presenteeism have changed markedly in the same period. In addition, the main interest in this study was to study the relationships between variables. Even if secular trends or sudden changes to health-related behaviors were to impact on levels of presenteeism in the population, there is little reason to believe that this would impact on the reported associations. In a similar vein, previous research has shown that while skewed samples represent a threat to the validity of prevalence estimates, associations between predictors and substance use outcomes are less prone to be affected [37]. Future studies with time series designs may investigate whether associations with alcohol-related outcomes, such as presenteeism, vary across time.

We are aware of differing conceptualizations of presenteeism among scholars. While some conceptualize presenteeism simply as the act of “*showing up for work even when one is ill*” ([38], p. 519), others have emphasized that de facto productivity loss is inherent in the concept of presenteeism, i.e., “*decreased on-the-job performance due to health problems*” ([1], p. 503). Since attending work despite having a health condition does not necessarily involve productivity impairments, and since attending work while ill, in an organizational perspective, primarily becomes of interest when work performance is impaired, we chose to conceptualize alcohol-related presenteeism as impaired work performance attributable to alcohol consumption.

## 5. Conclusions

The aim of this study was to explore associations between demanding job situations and alcohol-related presenteeism, with particular focus on variables derived from the JDC and ERI models. Employee overcommitment was found to increase the odds of reporting some degree of alcohol-related presenteeism during the last week. In addition, several sociodemographic variables were consistently associated with alcohol-related presenteeism across all tested statistical models. Based on this study, occupational health services and employers should especially focus on overcommitted employees when designing workplace health promotion programs. Modifying attitudes towards alcohol-related presenteeism among overcommitted employees may be of importance for safety at work. 

## Figures and Tables

**Table 1 ijerph-18-06169-t001:** Characteristics of the study sample and comparisons between participants reporting presenteeism versus no presenteeism.

Variables	Sample *n* = 6620	No Presenteeism *n* = 6147 (92.9%)	Presenteeism *n* = 473 (7.1%)	*p*
Age (Mean [SD])	45.0 (11.3)	44.9 (11.3)	46.0 (10.8)	0.04
Gender (*n* [%])				
Male	2003 (30.3)	1831 (91.4)	172 (8.6)	<0.01
Female	4617 (69.7)	4316 (93.5)	301 (6.5)	
Education level (*n* [%])				
Higher education	5129 (77.5)	4722 (92.1)	407 (7.9)	<0.001
High school education or lower	1491 (22.5)	1425 (95.6)	66 (4.4)	
Managerial responsibility (*n* [%])				
Top executive	143 (2.2)	130 (90.9)	13 (9.1)	0.06
Middle management	1161 (17.5)	1061 (91.4)	100 (8.6)	
Regular employee	5316 (80.3)	4956 (93.2)	360 (6.8)	
Job Demand Control model concepts (Mean [SD])				
Decision latitude	27.6 (3.7)	27.6 (3.7)	27.7 (3.9)	0.72
Psychological demands	12.9 (2.4)	12.9 (2.4)	13.0 (2.5)	0.48
Social support	25.4 (3.7)	25.5 (3.7)	25.3 (3.6)	0.49
Effort-Reward model concepts (Mean [SD])				
Effort	8.3 (1.8)	8.3 (1.9)	8.4 (1.8)	0.53
Reward	18.8 (2.7)	18.8 (2.7)	18.8 (2.7)	0.48
Overcommitment	13.8 (3.3)	13.7 (3.3)	14.2 (3.2)	<0.01
Effort-Reward Imbalance ratio	1.06 (0.33)	1.06 (0.32)	1.08 (0.35)	0.30
Risky drinking (*n* [%])				
AUDIT score ≥ 8 (risk)	750 (11.3)	631 (84.1)	119 (15.9)	<0.001
AUDIT score < 8 (no/low risk)	5870 (88.7)	5516 (94.0)	354 (6.0)	

Note. Statistical tests are independent *t*-tests (continuous variables) and Chi-square tests (categorical variables). AUDIT: Alcohol Use Disorders Identification Test.

**Table 2 ijerph-18-06169-t002:** Unadjusted associations with alcohol-related presenteeism (*n* = 6620).

Independent Variables	OR	95% CI
Age increase in 10 years	1.09 *	1.00–1.19
Gender	0.74 **	0.61–0.90
Education level	1.86 ***	1.43–2.43
Managerial responsibility	1.31 *	1.05–1.63
Decision latitude	1.01 ^ns^	0.98–1.03
Psychological demands	1.01 ^ns^	0.98–1.05
Social support	0.99 ^ns^	0.97–1.02
Effort	1.02 ^ns^	0.97–1.07
Reward	0.99 ^ns^	0.95–1.02
Overcommitment	1.04 **	1.01–1.07
ERI ratio	1.16 ^ns^	0.88–1.53
High-strain job	0.90 ^ns^	0.70–1.15
Low support	1.03 ^ns^	0.86–1.25
High overcommitment	1.25 *	1.04–1.52
High ERI ratio	1.01 ^ns^	0.82–1.24
Cumulative risk	1.04 ^ns^	0.96–1.13

Note. ERI is effort-reward imbalance ratio. Reference values are male gender, job type other than high-strain (i.e., low-strain, passive or active job), high support (above median), low ERI ratio (below median) and low overcommitment (below median). For all continuous variables, reference categories are lower values. Cumulative risk indicates number of categorical risk factors (high-strain job, low support, high ERI ratio, and high overcommitment), where the score range is 0–4. * *p* < 0.05; ** *p* < 0.01; *** *p* < 0.001; ns = non-significant (*p* ≥ 0.05).

**Table 3 ijerph-18-06169-t003:** Associations between alcohol-related presenteeism and sociodemographic variables, Job Demand Control model variables, Effort-Reward Imbalance model variables, work environment risk factors and cumulative risk (*n* = 6620).

Independent Variables	Model 1 OR (95% CI)	Model 2 OR (95% CI)	Model 3 OR (95% CI)	Model 4 OR (95% CI)	Model 5 OR (95% CI)	Model 6 OR (95% CI)
Age increase in 10 years	1.10 * (1.01–1.20)	1.10 * (1.01–1.20)	1.10 * (1.01–1.20)	1.10 * (1.01–1.20)	1.10 * (1.01–1.19)	1.10 * (1.01–1.20)
Gender	0.75 ** (0.62–0.92)	0.75 ** (0.62–0.92)	0.74 ** (0.61–0.90)	0.75 ** (0.61–0.91)	0.74 ** (0.61–0.91)	0.75 ** (0.61–0.91)
Education	1.89 *** (1.44–2.47)	1.94 *** (1.47–2.55)	1.89 *** (1.44–2.48)	1.88 *** (1.44–2.47)	1.85 *** (1.41–2.43)	1.89 *** (1.44–2.47)
Managerial responsibility	1.13 ^ns^ (0.90–1.42)	1.15 ^ns^ (0.91–1.46)	1.14 ^ns^ (0.90–1.44)	1.14 ^ns^ (0.86–1.52)	1.09 ^ns^ (0.87–1.38)	1.12 ^ns^ (0.96–1.41)
Decision latitude	-	0.99 ^ns^ (0.96–1.02)	-	-	-	-
Psychological demands	-	1.01 ^ns^ (0.97–1.05)	-	-	-	-
Social support	-	1.00 ^ns^ (0.97–1.02)	-	-	-	-
Effort	-	-	0.96 ^ns^ (0.90–1.02)	-	-	-
Reward	-	-	0.98 ^ns^ (0.95–1.02)	-	-	-
Overcommitment	-	-	1.04 * (1.01–1.08)	-	-	-
ERI ratio	-	-	-	0.85 ^ns^ (0.56–1.27)	-	-
High-strain job	-	-	-	-	0.89 ^ns^ (0.68–1.16)	-
Low support	-	-	-	-	1.04 ^ns^ (0.86–1.26)	-
High overcommitment	-	-	-	-	1.25 * (1.02–1.53)	-
High ERI ratio	-	-	-	-	0.96 ^ns^ (0.76–1.22)	-
Cumulative risk	-	-	-	-	-	1.04 ^ns^ (0.96–1.13)
*Model parameters*						
Model χ^2^	39.4 ***	41.0 ***	47.2 ***	40.2 ***	44.7 ***	40.2 ***
Nagelkerke R^2^	0.02	0.02	0.02	0.02	0.02	0.02
Cox and Snell R^2^	0.01	0.01	0.01	0.01	0.01	0.01

Note. ERI ratio is effort-reward imbalance ratio. Cumulative risk indicates number of categorical risk factors (high-strain job, low support, high ERI ratio, and high overcommitment), where the score range is 0–4. Reference values are male gender, job type other than high-strain job, high support (above median), low ERI ratio (below median) low overcommitment (below median). For continuous variables, reference categories are lower values. Model 1 shows associations between the sociodemographic variables and presenteeism. Models 2–6 show associations with presenteeism for Job Demands Control model variables, Effort-Reward Imbalance model variables, effort-reward ratio, work environment risk factors and accumulated risk, respectively. All associations are adjusted for the sociodemographic variables. * *p* < 0.05; ** *p* < 0.01; *** *p* < 0.001; ns = non-significant (*p* ≥ 0.05).

## Data Availability

The data from the study contain potentially sensitive information. In accordance with restrictions imposed by the Regional Committees for Medical and Health Research in Norway (approval no. 2014/647), data must be stored on a secure server at the University of Stavanger. The contents of the ethics committee’s approval resolution as well as the wording of participants’ written consent do not render open public data access possible. Access to the study’s minimal and depersonalized data set may be requested by contacting the Faculty of Health Sciences at University of Stavanger (post@uis.no).

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
