# Peer review of "Are Demanding Job Situations Associated with Alcohol-Related Presenteeism? The WIRUS-Screening Study"

_ijerph, 2021, doi:10.3390/ijerph18116169_

Round 1

Reviewer 1 Report

I suggest to accept the manuscript in the revised form

Author Response

Thank you.

Reviewer 2 Report

Dear authors,

You have conducted research on presenteeism related to psychosocial and workplace risk factors. The analyzes carried out are pertinent and the information provided allows the study to be reproduced.

Your manuscript is very interesting but I need you to answer some questions:

METHODS

Design:

The authors say they collect cross-sectional data but do so between 2014-2019. Wouldn't it be a longitudinal study? The authors should clarify this point and specify the research design.

You did not consider between 2014-2019 to carry out a longitudinal investigation?

Sample and inclusion criteria

The response rate should be done when incomplete information is discarded.

Did you take into account whether the employees were permanent or temporary?

REFERENCES

Some references do not meet the journal guidelines. The authors have mixed APA and Vancouver citation standards.

Some references are incomplete or have errors. The authors should review this section.

Author Response

Authors: Thank you for the comments. We have addressed each comment point by point below, and all changes in the manuscript have been performed using track changes for word. We look forward to hearing from you.

***************************************************************************

R2: You have conducted research on presenteeism related to psychosocial and workplace risk factors. The analyzes carried out are pertinent and the information provided allows the study to be reproduced. Your manuscript is very interesting but I need you to answer some questions: Design: The authors say they collect cross-sectional data but do so between 2014-2019. Wouldn't it be a longitudinal study? The authors should clarify this point and specify the research design.

Authors: This cross-sectional study collected data only once from each of the participants. However, we have modified our description of the study design; see section 2.1.

R2: You did not consider between 2014-2019 to carry out a longitudinal investigation?

Authors: A longitudinal investigation could obviously be interesting to track employees’ patterns of drinking over time as they progress (or stagnate) in their career, but that goes beyond the scope of the present study. The present study is part of a larger study where the main purpose was to screen employees for risky alcohol use, and then invite individuals classified as ‘risky drinkers’ to an intervention. Thus, a cross-sectional study was appropriate for the screening purpose. 

R2: Sample and inclusion criteria: The response rate should be done when incomplete information is discarded.

Authors: While we would maintain that responders may not have completed every question in the questionnaire, we have included information about the proportion of eligible participants included in the analysis for this study; see section 2.2.

R2: Did you take into account whether the employees were permanent or temporary?

Authors: No, according to inclusion criterion #2 (see section 2.2), ‘being employed’ did not differentiate between employees in permanent versus temporary positions. Therefore, we do not have data on this issue, and we have included this as a point in the study limitations.

R2: Some references do not meet the journal guidelines. The authors have mixed APA and Vancouver citation standards. Some references are incomplete or have errors. The authors should review this section.

Authors: We have reviewed in-text citations and the reference list, and we are unable to find any deviation from the journal’s reference guidelines. However, if we have failed to spot any mistakes, we apologize in advance and are happy to revise them if needed.

This manuscript is a resubmission of an earlier submission. The following is a list of the peer review reports and author responses from that submission.

Round 1

Reviewer 1 Report

This is an interesting work on a relevant public health topic.

A few comments:

  1. The regression is correlational in nature. So, it is possible that overcommitment contributes to lead to higher presenteism, as noted by the authors, but it is also possible, for example, that individuals with higher presenteism (i.e. individuals with higher degree of alcohol use) have an unbalanced perception of their work duties and thus perceive higher levels of overcommitment. This potential bidirectional relationhip should be mentioned in the discussion.
  2. In table 1, what the numbers in parentheses indicate in relation to the lines on AUDIT scores?
  3. As the study is based on logistic regression analyses, can the authors state whether they checked that the data meet the necessary assumptions for such analyses?

Author Response

Authors: We thank the reviewers for their constructive comments. All comments have been addressed point by point in the response letter, and all changes have been performed using track changes. We look forward to hearing from you.

***************************************************************************

Reviewer 1 (R1): This is an interesting work on a relevant public health topic. A few comments: The regression is correlational in nature. So, it is possible that overcommitment contributes to lead to higher presenteism, as noted by the authors, but it is also possible, for example, that individuals with higher presenteism (i.e. individuals with higher degree of alcohol use) have an unbalanced perception of their work duties and thus perceive higher levels of overcommitment. This potential bidirectional relationhip should be mentioned in the discussion.

Authors: We agree, and the possibility of reversed causality is now discussed in more detail in the study limitations section.

R1: In table 1, what the numbers in parentheses indicate in relation to the lines on AUDIT scores?

Authors: The numbers in parentheses indicate the proportion of individuals within the relevant category. E.g., a total of 750 individuals were classified as risky drinkers (AUDIT score of 8 or above). Of these, 119 (15.9%) indicated some degree of alcohol-related presenteeism.

R1: As the study is based on logistic regression analyses, can the authors state whether they checked that the data meet the necessary assumptions for such analyses?

Authors: Binary logistic regression analysis requires a categorical dependent variable, independent observations, independent variables linearly associated with log-odds, little or no multicollinearity between the independent variables, and a generally large sample size. All these requirements were met. After transforming the independent variables using the natural logarithmic function, none of the transformed variables were significantly associated with presenteeism (Box-Tidwell test). The strongest bivariate correlation (Pearson’s r) between pairs of independent variables was shown for efforts and demands (0.69). Some relatively strong associations (about 0.50) were also shown for pairs of variables expected to be associated (such as efforts and overcommitment [0.50] and rewards and support [0.45]). Otherwise, associations were mostly small to moderate (0.30 or below). See added information in the data analysis section.

Reviewer 2 Report

Dear authors,

This article analyzes the alcohol-related hunch in workers of public and private companies. The article has a large sample and is interesting. The structure and methodology seem correct. The results are presented in a clear and easy to understand way.

However, I need you to correct and explain some questions to me:

MATERIALS AND METHODS

Design

The authors must specify the research design. Collecting cross-sectional data over 5 years is not very clear. The authors should explain this better.

Sample:

How was the sample chosen? Authors should better specify the inclusion criteria.

The authors must include the response rate of the participants in the study.

REFERENCES

Many bibliographies are obsolete. The bibliographic citations used are more than 5 years old (51.8 %). The authors must update and arrange the bibliography.

Author Response

Authors: We thank the reviewers for their constructive comments. All comments have been addressed point by point in the response letter, and all changes have been performed using track changes. We look forward to hearing from you.

***************************************************************************

Reviewer 2 (R2): This article analyzes the alcohol-related hunch in workers of public and private companies. The article has a large sample and is interesting. The structure and methodology seem correct. The results are presented in a clear and easy to understand way. However, I need you to correct and explain some questions to me: Design: The authors must specify the research design. Collecting cross-sectional data over 5 years is not very clear. The authors should explain this better.

Authors: The reason for the substantial time used to collect the data is stated in the design section, whereas possible consequences are discussed in the study limitations section.

R2: Sample: How was the sample chosen? Authors should better specify the inclusion criteria. The authors must include the response rate of the participants in the study.

Authors: The inclusion criteria and response rate have been specified, see revised ‘Sample and inclusion criteria’ section.

R2: Many bibliographies are obsolete. The bibliographic citations used are more than 5 years old (51.8 %). The authors must update and arrange the bibliography.

Authors: Most of the old citations refer to classical work important to our study (e.g., related to the theoretical models and derived measures used in the study), and we have retained these in the revised manuscript. However, we have included some recent references that provide relevant context for our study (see revised introduction and discussion sections).

Round 2

Reviewer 2 Report

Dear authors,

A cross-sectional study lasting 5 years is rare. I have asked him to explain more details to us. I don't know what kind of companies they are, I don't know what kind of workers, I don't know what functions they have ...

You have not specified how you sampled the total population.

References remain out of date with the same percentage. 

Best regards